



# A historical reconstruction of cropland in China from 1900 to 2016

Zhen Yu[1], Xiaobin Jin[2], Lijuan Miao[3], Xuhong Yang[2]

[1]Key Laboratory of Agrometeorology of Jiangsu Province, Institute of Ecology, School of Applied Meteorology, Nanjing University of Information Science and Technology, Nanjing, 210044, China
[2]School of Geography and Ocean Science, Nanjing University, Nanjing, 210023, China
[3]School of Geographical Sciences, Nanjing University of Information Science and Technology, Nanjing, 210044, China

*Correspondence to*: Zhen Yu (zyu@nuist.edu.cn)

**Abstract.** A spatially-explicit cropland distribution time-series dataset is the basis for the accurate assessment of
biogeochemical processes in terrestrial ecosystems and their feedback to the climate system; however, this type of dataset is lacking in China. Existing cropland maps have a coarse resolution, are intermittently covered, or the data are inconsistent. We reconstructed a continuously covered cropland distribution dataset in China spanning from 1900 to 2016 by assimilating multiple data sources. In total, national cropland acreage expanded from 77.72 Mha in 1900 to the peak of 151.00 Mha in 1979, but it consistently decreased thereafter to 134.92 Mha in 2016. The cropland was primarily distributed in three historically
cultivated plains in China: the Sichuan Plain, the Northern China Plain, and the Northeast China Plain. Cropland abandonment was approximately 29.90 Mha; it was mainly concentrated in the Northern China Plain and the Sichuan Plain and occurred during the 1990–2010 period. Cropland expansion was over 74.30 Mha; it was primarily found in the southeast, northern central, and northeast regions of China and occurred before 1950. In comparison, the national total and spatial-distribution of cropland in the Food and Agriculture Organization (FAO) of the United Nations and the History Database of the Global
Environment (HYDE) were distorted during the period of 1960–1980 due to the biased signal from the Chinese Agricultural Yearbook. We advocate that newly reconstructed cropland data, in which the bias has been corrected, should be used as the updated data for regional and global assessments, such as greenhouse gas emission accountings and food production simulations. The cropland dataset is available via an open-data repository (https://doi.org/10.6084/m9.figshare.13356680) (Yu et al., 2020).

**1 Introduction**

Land use and cover change (LUCC) has transformed over 1/3 of the planet's surface, altering regional and global climate via changes in biogeochemical and biogeophysical processes (Foley et al., 2005; Goldewijk et al., 2017). As the dominant drivers of LUCC, agricultural activities that produce food, fiber, and livestock have triggered far-ranging consequences, such as soil erosion, nutrient depletion, desertification, salinization, and acidification (FAO and ITPS, 2015; Keesstra et al., 2016;
Sanderman et al., 2017). It has been reported that cropland encroachment into natural lands could alter the land-atmosphere exchange of energy and water, thus impacting temperature, humidity, precipitation, convection, and wind (Kueppers et al.,

2007). On the one hand, agricultural plants have unique biophysical characteristics that are different from natural vegetation, directly influencing local, regional, and global climate systems. On the other hand, human domination of croplands has also transformed the hydrologic and nutrient cycles through drainage, irrigation, and fertilization (Carlson et al., 2017; Castellano

et al., 2019). Thus, a better understanding of spatial-temporal conversions in cropland will greatly benefit the quantification of national and global carbon budgets, carbon-climate feedback, and other land-based ecosystem functions (Yu and Lu, 2018).

During the long history of crop cultivation in China, dating back more than 7000 years, 130–176 Mha of natural lands were converted for food production (FAO, 2018; Lai et al., 2016). The land use change process has under gone unprecedented transformation due to the population boom in China, which has tripled since 1900, during which the growing demand for food

and fiber has expanded agricultural activities into vast and previously undisturbed lands. Nevertheless, ineffective agricultural technology, combined with improper planning and policy failures, has deteriorated the natural ecosystem and contributed to intensive greenhouse gas (GHG) emissions. Thus, a long-term and spatially-explicit cropland distribution dataset will be the foundation for an accurate assessment of the land-atmosphere interactions.

The complex cropland change in China is influenced by the country's economics, politics, culture, and topography.

A typical challenge is the scarce data records, which is further aggravated by the lack of uniform standards in data surveys, and the changes in administrative boundaries that occurred before the 1960s. Many studies have focused on reconstructing historical cropland distribution in China and have produced cropland maps for a few intermittent years (Li et al., 2015, 2018; Lin et al., 2009; Liu and Tian, 2010; Wei et al., 2019; Yang et al., 2016). Among all these datasets, Yang et al. (2015a, 2015b) is prominent in providing a reliable and relatively high resolution (1 km) of croplands in historical years using a "bottom-up"

model equipped with the Cellular Automaton approach. Most of these products describe cropland in Boolean maps, namely each grid cell is either completely occupied by cropland or other non-crop vegetation. This approach is more suitable for mid- and high-resolution images, but a large bias will accumulate during upscaling (Yu and Lu, 2018). This has been evidenced in a previous study, which reported that abandoned croplands have been underestimated by up to 40% during aggregation in low-resolution gridded maps (Zumkehr and Campbell, 2013).

Another approach is to use fractional coverage to describe cropland distribution in each grid cell, which is common for mid- and low-resolution gridded maps, such as the History Database of the Global Environment (HYDE v3.2) (Goldewijk et al., 2017). HYDE has been widely applied in global and regional model simulations due to its unique advantage in depicting cropland distribution with long-term (10,000 B.C. to 2017 A.D.) and global coverages (e.g., Wang et al., 2020). Nonetheless, the relatively coarse resolution (5 arc-min) has restricted its applicability. Recent studies have produced high resolution (30

m–300 m) cropland maps with the aid of a computer and increasingly abundant satellite images (e.g., ESA CCI, Globalcrop30m). However, all these products are either inconsistent with the official cropland data (e.g., ESA CCI, MODIS) or are not continuous (e.g., Globalcrop30m in 2015, GlobeLand30 in 2000, 2010, and 2020).

To better serve the economics plans developed since 1949, data on cropland acreages have been collected under increasingly organized protocols in China. Since that time, the longest continuous records are from the Chinese Statistical

Yearbook (CSY); however, the accuracy of those records has been questioned (Bi and Zheng, 2000; Feng et al., 2005; Zheng,



1991). Instead, three other critical surveys have been widely acknowledged as being relatively reliable in representing cropland areas in China: the national and official cropland estimation in 1953 (108.53 Mha, Ministry of Finance of the People's Republic of China), the second national cropland estimation (130.04 Mha), known as the First National Land Survey (FNLS), from 1985 to 1996, and the Second National Land Survey (SNLS) from 2007 to 2009 (135.38 Mha). However, the information obtained

from these surveys is still inconsistent in comparison to annual land use change records (Fig. S1). There is an urgent need to elucidate the discrepancies between these datasets and to build a temporal consistent cropland acreage data-series for reconstructing spatially-explicit cropland maps.

In this study, we used a simple but straightforward approach to reconstruct the annual cropland density maps by harmonizing long-term tabular data and gridded images covering the period of 1900 to 2016. Our objectives are to: 1) develop

a continuous dataset depicting the cropland distribution in China by reconciling accuracy, temporal coverage, and spatial resolutions among different data sources; and 2) examine the distribution of cropland expansion and abandonment using the produced dataset. Cropland distribution and conversion are characterized by cropland density in each 5 km-by-5 km grid cell.

## 2 Data and methods

### 2.1 Study area

This study focused on mainland China (excluding Taiwan, Hong Kong, and Macaw), which is also one of the most intensively cultivated areas in the world. Although China's administrative boundary has changed a few times since 1900, the official statistical records were generally consistent after the founding of the People's Republic of China in 1949. Thus, the data used before 1949 were summarized according to the administrative boundary defined by Yang et al. (2015b), while the most recent administrative boundary was used for the period after 1949.

### 2.2 Tabular data

It is necessary to obtain reliable cropland area time-series data before a cropland distribution map can be reconstructed using the model we developed previously (Yu and Lu, 2018). Here, data from four tabular datasets and some intermittently reported data that specifically focus on mainland China were obtained, compiled, and adjusted covering the period of 1949 to 2018 (Table 1). The first statistical dataset is from the National Land and Resources Bulletin (NLRB) provided by the Ministry of

Land and Resources of China covering the period of 2001–2017. The NLBR has continuously updated the national-level cropland area since 2001; it is also the most authoritative report officially released by the Chinese government.

The second statistical dataset is the Land and Resources Statistical Yearbook (CLRSY) published by the Ministry of Land and Resources of China covering the period of 1996 to 2018; the CLRSY data from 2001 to 2018 were either missing or the same as the NLRB data. Therefore, the inter-annual changes of national cropland area documented in CLRSY for the

period 1996 to 2000 were used to extend the NLRB data back to 1996.





The third statistical dataset is the Chinese Agricultural Yearbook (CAY) provided by National Bureau of Statistics of China, which documented the annual cropland area from 1980 to 2018 at the provincial level; however, approximately 19% of the records are missing. Thus, the CAY data were gap-filled using the linear interpolation approach and further adjusted for each province (Fig. S2). The CAY helps extend the data on cropland acreages back to 1980.

100       The fourth statistical dataset is the national crop production from 1949–2016 obtained from the CSY. The crop production data, combined with intermittently reported cropland acreage data from other studies, were used to rebuild the cropland area between 1949 and 1979 (Fig. S3, Table S1).

Table 1. Data sources of the cropland inventory.

| Datasets | Year | Variable | Adjustment made in this study | Sources |
|---|---|---|---|---|
| National Land and Resources Bulletin (NLRB) | 2007–2017 | National | National-level cropland area data from 2009 to 2016 were used as the baseline; the national cropland area data of 2001–2008 were adjusted. | The Ministry of Land and Resources of China |
| China Land and Resources Statistical Yearbook (CLRSY) | 1999–2001 | National | National-level cropland area data adjusted for 1996–2000 | The Ministry of Land and Resources of China |
| China Agricultural Yearbook (CAY) | 1981–2018 | Provincial | Inter-annual cropland changes at the provincial level used after gap-filling and adjustment. | National Bureau of Statistics of China |
| Chinese Statistical Yearbook (CSY) | 1949–2016 | National | Crop production data used for cropland area reconstruction for 1949–1979. | National Bureau of Statistics of China |

## 105   2.3 Satellite-based image products

Multiple sources of remote sensing products were used in this study (Table 2). The products with the highest spatial resolution are at 30 m from: 1) the global cropland provided by Global Food Security Analysis-Support Data (GFSAD30) Project, 2) the China Land Use and Cover Change (CNLUCC) produced by the Institute of Geographic Sciences and Natural Resources Research, Chinese Academy of Sciences, 3) the GlobeLand30 developed by the Ministry of Natural Resources of China, and

4) FROM-GLC produced by Gong et al. (2013) from Tsinghua University. In comparison, the European Space Agency Climate Change Initiative (ESA CCI) Land Cover time-series and the MODIS Land Cover product (MCD12Q1 collection 6) are 300 m and 500 m resolution, but covers much longer periods, 1992–2018 and 2001–2019, respectively. The other four gridded datasets were prepared by Ran and Li (2006) at 1 km resolution and intermittently cover the period of 1980 to 2000. Additionally, the digitized 1:1,000,000 vegetation map produced by the Compiling Committee of the Vegetation Maps of China (CCVM) (Hou, 2001)) is also rasterized to provide cropland information from the 1980s.



Among these datasets, GFSAD30 is specifically designed by using multitemporal Landsat imagery for 3 years (2013–2015), over 100,000 reference data samples, and other auxiliary data sources. Similarly, the CNLUCC was also developed from Landsat TM images with supplemental data from the Huanjing-1 satellite. The CNLUCC has been intensively interpreted using the human-computer interactive approach, specifically focusing on China, which intermittently covers the period of

1980s to 2018 at 30 m resolution (Liu et al., 2014). The dataset has been rigorously validated and widely applied for land use change analyses in China (Liu et al., 2003, 2010, 2014, 2002). The GlobeLand30 maps were produced from multiple sources of images by the Ministry of Natural Resources of China, including TM5 ETM+ and OLI multispectral images of Landsat and HJ-1 (China Environment and Disaster Reduction Satellite), the 16-meter resolution GF-1 (China High Resolution Satellite) multispectral image. The GlobeLand30 have also been intensively validated with an accuracy of over 83% (Chen et al., 2016).

Similar, FROM-GLC images were extracted from Landsat and validated with an approximate accuracy of 71% (Gong et al., 2013; Li et al., 2017). Therefore, in the present study, GFSAD30, CNLUCC, FROM-GLC, and GlobeLand30 are the major data sources for cropland distribution reconstruction that has occurred since the 1980s.

Table 2. Gridded data collected for cropland reconstruction.

| Datasets | Year | Resolution/scale | Sources |
|---|---|---|---|
| MODIS MCD12Q1 | 2001–2019 | 500 m | The National Aeronautics and Space Administration |
| ESA CCI | 1992–2018 | 300 m | European Space Agency |
| Global Cropland 30m (Globalcrop30m) | 2015 | 30 m | Global Food Security Analysis-Support Data at 30 Meters (GFSAD30) Project |
| IGBP Data and Information System | 1992–1993 | 1 km | Loveland et al. (2000) |
| UMD Land Cover | 1992–1993 | 1 km | Hansen et al. (2000) |
| GLC2000 | 2000 | 1 km | Bartholomé and Belward (2005) |
| CAS1990 | 1980s–1990s | 1 km | Ran and Li (2006) |
| WESTDC Land Cover Product v2.0 | 2000 | 1 km | Ran (2013) |
| Vegetation map | 1980s | 1:1,000,000 | CCVM* |
| CNLUCC** | 2018, 2015, 2010, 2008, 2005, 2000, 1995, 1980s | 30 m | RESDC*** |
| FROM-GLC | 2010, 2015, 2017 | 30 m | Gong et al. (2019) |
| GlobeLand30 | 2000, 2010, 2020 | 30 m | Chen et al. (2016); Jun et al. (2014) |

*CCVM: The Compiling Committee of the Vegetation Maps of China; **CNLUCC: China Land Use and Cover Change;

***RESDC: The Data Center for Resources and Environmental Sciences, Chinese Academy of Sciences (RESDC) (http://www/resdc.cn).



### 2.4 Approach for reconstructing historical cropland acreage data at the national and provincial levels

Cropland area is defined as the area of land that has been cultivated with plants, including fallow cropland. However, discrepancies in the reported national cropland areas were reported in previous studies as well as in officially released time-series datasets (Fig. S1). Despite the large differences, the cropland areas reported by different official agencies were generally consistent in the approaches that were used for data collection in each specific period. Thus, we made two assumptions about using the collected time-series datasets: 1) although the officially released cropland acreage datasets are systematically biased, the inter-annual variations are reliable; and 2) with the development of technology, cropland areas and the reported changes are increasingly reliable. Based on these assumptions, we retrospectively reconstructed information about the cropland areas year-by-year using tabular data and gridded maps from different sources. In this study, the officially released NLRB report in 2017 (cropland area in 2016) is used as the benchmark data, which is also the most recent and authoritative record of provincial cropland acreage available. Therefore, the annual amount of national cropland areas during historical years were reconstructed by adjusting the benchmark year data using inter-annual cropland change information derived from different sources, which can be divided into six periods (Fig. 1).

The first period is covered by the annual NLRB reports (2001–2017). In the NLRB, the cropland acreage has an abrupt change from 2008 to 2009, due to the systematic bias that was corrected using the SNLS in 2009 (Fig. S1). Specifically, the SNLS was launched in 2007 using state-of-the-art technologies to delineate the use of each parcel unit of the land in China. The two-year survey built the first comprehensive database covering land use information from the county level to the national level with the aid of various airborne and spaceborne remote sensing platforms and field investigations. Therefore, we believe that the updated NLRB reports (2009–2017) are more accurate and reliable than the reports from 2001 to 2008. Apparently, the cropland areas have been systematically and substantially underestimated by 13–14 Mha during the period of 2001 to 2008 (Fig. S1). Thus, instead of using the original reported number, the inter-annual cropland changes from the annual NLRB reports were adopted to extend the data back to 2001. Note that the SNLS map is not accessible for the public; other contemporaneous maps with commensurate quality from multiple sources were alternatively used for spatial mapping (e.g., GFSAD30, GlobeLand30).

The second period ranges from 1996 to 2000 with cropland data obtained from the annual CLRSY released by the Ministry of Land and Resources of China. These officially reported cropland data were actually annually updated from baseline year data in 1996 and the land use change areas reported by local government. Similarly, we believe the cropland area data from the baseline year in 1996 are systematically biased, while the inter-annual changes in the cropland area during the period of 1996 to 2000 were reliable and were used to extend the cropland data back to 1996.

The third period is from 1986 to 1995, during which the inter-annual changes in the cropland area were provided by the Ministry of Land and Resources of China. Similar to the 1996–2000 period, the land use changes were reported from local government and were used to extend the cropland area data back to 1986 in this study.



The fourth period ranges from 1980 to 1985, during which the national cropland area changes were derived from the

gap-filled data of the CAY. The CAY data were also officially and annually updated from the cropland changes collected by the local government in each province. This helped extend the national cropland data back to 1980.

The fifth period ranges from 1949 to 1979, the data from which were specifically rebuilt using the crop production-area relationship derived from key years intermittently documented from 1949 to 1960. According to Feng et al. (2005), the cropland area data officially reported before 1960 were relatively reliable, yet the data during the period of 1960 to 1980 were

distorted due to political issues (see Section 4.1). Similar to Feng et al. (2005), we found a high correlation between cropland area and crop production using records collected before 1960 ($R^2 = 0.92$; Fig. S3, Table S1). Based on the built relationship, we then interpolated the cropland area data for 1949–1980, and extracted the inter-annual changes to extend the cropland time-series data back to 1949.

For the last period of 1900 to 1948, we used the trends extracted from Yang et al.'s (2015b) cropland data from 1887,

1933, and 1952.

Similar to the national cropland area data reconstruction, the provincial cropland area data reconstruction can also be divided into three periods. The first period is from 2013 to 2016, during which data on the provincial cropland areas were directly provided in the NLRB reports. For the periods before 2013, the provincial cropland areas were adjusted using cropland area data from 2013 as the baseline. For example, during the second period of 1980 to 2012, we first gap-filled the cropland

area data in each province using linear interpolation (Fig. S2), and then we adjusted the inter-annual variations by removing abrupt changes (Fig. S2). Due to the lack of provincial data before 1980, we proportionally adjusted the amount of cropland area in each administrative boundary using national acreage data.

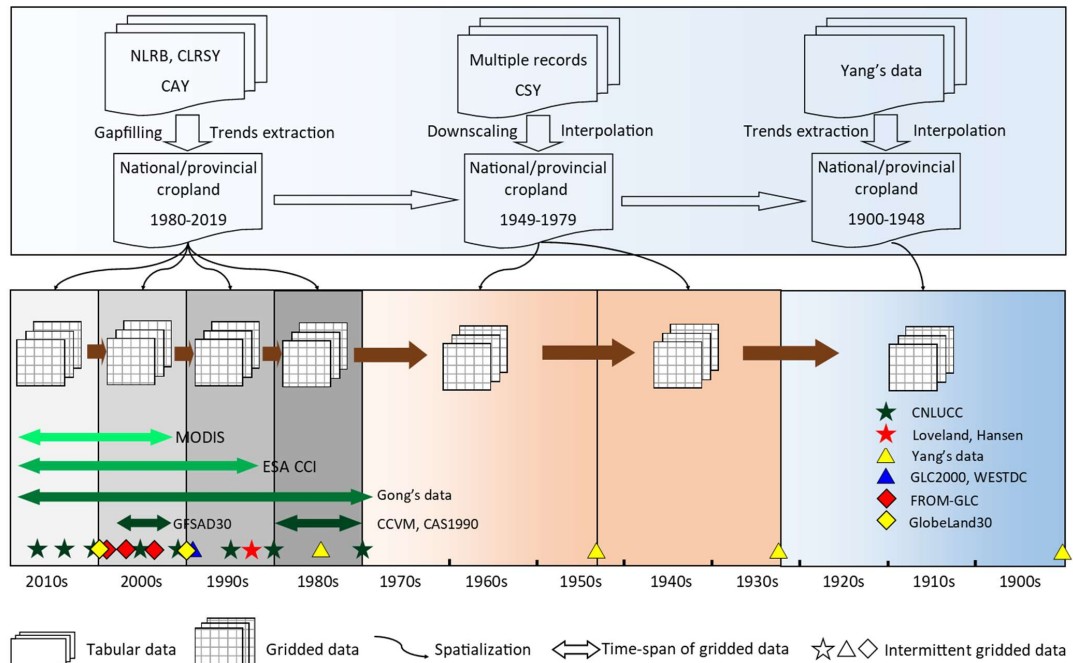

Figure 1. Methodology flow chart and datasets used in the cropland reconstruction. Upper boxes are the tabular survey data and the lower boxes are the gridded images. (CAY: Chinese Agricultural Yearbook; CSY: Chinese Statistical Yearbook; NLRB: National Land and Resources Bulletin; CLRSY: China Land and Resources Statistical Yearbook; MODIS: Moderate Resolution Imaging Spectroradiometer Land Cover Type Collection 6 (MCD12Q1); ESA CCI: the European Space Agency (ESA) Climate Change Initiative Land Cover; Gong et al.'s data: human settlement (urban and rural) data (Gong et al., 2013); GFSAD30: Global Food Security Analysis-Support Data at 30 meters; CCVM: 1:1,000,000 vegetation map of China; CAS1990: Land Cover data originated from 1:1,000,000 land use map produced by the Institute of Geographic Sciences and Natural Resources Research, Chinese Academy of Sciences; CNLUCC: China Land Use and Cover Change; Loveland: Land cover map developed by Loveland et al. (2000); Hansen: Land cover map produced by Hansen et al. (2000); Yang's data: cropland data from Yang et al. (2015); GLC2000: Global Land Cover 2000 database from the Joint Research Centre of European Commission; WESTDC: A land cover map compiled by the Chinese Academy of Sciences and edited by Ran and Li (2013); FROM-GLC: land cover maps produced by Gong et al. (2013) from Tsinghua University; GlobeLnad30: land cover maps produced by the Ministry of Natural Resources of China; the map reconstructed in one year was used as the initial condition for the following year's cropland reconstruction).



### 2.5 Approach for spatializing cropland distribution in China

Based on the provincial cropland acreage reconstruction, we spatialized the cropland distribution using gridded maps as
ancillary data. More specifically, we implemented the model we previously developed to allocate cropland spatially depending
on the cropland potential (possibility) (Yu and Lu, 2018). Here, we made the following two assumptions in constructing the
cropland maps: 1) the national and provincial cropland areas derived from the officially released NLRB reports are more
reliable than the gridded maps produced by other institutions; and 2) each gridded product provides potential distribution
(possibility) of cropland but suffered from different sources of errors. This is understandable as the officially released reports
were based on the SNLS, which is, to date, the most intensive, sophisticated, and authoritative survey conducted in China.
Thus, the NLRB statistics were given the highest priority to confine the cropland area in each province, while other gridded
products were used for spatial allocation of the croplands as they were less intensively validated and were less accurate.

Based on the above assumptions, we constructed cropland maps for each of the periods as shown in Fig. 1. Generally,
the idea is to allocate a prescribed cropland area to each province with priority given to gird cells that have a higher possibility.
Thus, we constructed a potential cropland map using the gridded images available during each of the periods under study. To
do so, three types of gridded images were directly resampled to 100 m for analysis. The first type of gridded images was
obtained from the remote sensing products, which were converted to Boolean type and assigned weighting factors with a higher
value given to a higher resolution image. The second type of gridded images is the weighted map created using urban and rural
maps produced by Gong et al. (2019). The weighted values are the distance to the nearest urban and rural area for each grid
cell, assuming that land close to human settlements were prone to be cultivated, while lands far from settlements have higher
tendency to be abandoned. The third type is the cropland constructed in each of the previous years, which assumes that the
cropland area data between years are correlated. The sum of these three types of weighted images were used as the potential
cropland map. For example, in reconstructing cropland maps for 2005, we first identified all the gridded products available
during the decade, including MODIS, ESA CCI, Global Cropland 30m, and CNLUCC maps. The four types of gridded images
were assigned weighting scores ranging from 1 to 4, with a higher score given to a higher resolution map. Moreover, the
reconstructed cropland in 2006 and the buildup map in 2003 by Gong et al. (2019) were used. A ranked potential map was
created by summing all the weighted images, which was then used to determine the cropland distribution for each province.
Specifically, for the pre-satellite era (1900 to 1979), the ranked map was created using the gridded map produced by Yang et
al. (2015b) and the earliest year data in each of the available gridded data series (e.g., MODIS LC in 2001, CNLUCC in the
1980s).

During this process, if the map-based cropland acreage of a province was higher than the inventory data, a certain
number of 100-m cropland grid cells would be removed to keep the total cropland area consistent with the reconstructed
national time-series data. The removal was prioritized to the low-ranking score grid cells. Otherwise, if the cropland area of a
province was higher than it was in the previous year, the discrepancy would be added (i.e., cropland abandoned) using the
ranked map with priority given to the high-ranking score grid cells. These processes trigger the occurrences of crop



abandonment and expansion events. After spatializing the cropland at 100 m gridded maps, we used a 5-km window to calculate the cropland percentage (Fig. 1).

After the 5 km × 5 km cropland maps were reconstructed, we compared our results with the findings reported in previously published studies and available gridded maps. Specifically, the widely used HYDE v3.2 (Goldewijk et al., 2017)
data were employed for spatial pattern comparisons of cropland distribution, abandonment, and expansion. Since HYDE is a prominent product advantaged in representing long-term cropland distribution with global coverage (Yu and Lu, 2018), the comparison with HYDE will be informative for future simulations and improvements. The cropland abandonment and expansion data from our reconstructed maps were derived from a grid-by-grid comparison of the 100-m cropland maps and presented here by up-scaling to 5 km. Therefore, the abandonment year and expansion year indicate the latest conversions of
cropland-to-noncropland and noncropland-to-cropland, respectively. Limited by the spatial resolution, HYDE-based cropland abandonment is the difference between the current maximum cropland coverage and that of 2016, while the expansion is the difference of cropland coverage between 1900 and 2016. Thus, the HYDE abandonment year is the time in which the cropland area began to decrease in each of the 5 arc-min grids. Specifically, the HYDE-based expansion year is defined here as the date of the largest cropland increment. Therefore, the HYDE-based cropland conversion date is an approximate representation
because it was not possible to obtain the exact occurrence event in each grid cell. This study seeks to correct the biases of historical cropland area data, reduce the uncertainty in cropland distribution, and provide an alternatively reliable estimate of long-term cropland maps at a relatively higher resolution than is possible with HYDE.

## 3 Results

### 3.1 Changes in the historical cropland area in China

Rebuilt national cropland acreage increased from 81.40 Mha in 1900 to the first peak of 124.36 Mha until the end of the 1950s. Since 1960, the national cropland area has rapidly increased to its maximum size (151.00 Mha) in the early 1980s, followed by a steady and consistent decline thereafter. We compared the national cropland area rebuilt in this study with other reports, surveys, and published results (Fig. 2). The time-series datasets are generally consistent in overall trends, although the inter-annual changes and amplitudes varied greatly between the studies. Our reconstructed cropland acreage data and HYDE data,
which are also the two datasets with full coverage of the entire study period, are consistent with trends seen before the 1930s and after the 1990s (Fig. 2). Conspicuously, the discrepancy between the two datasets increased to about 30–40 Mha from the 1940s to the 1980s. Note that HYDE was built using FAO statistics; thus, the trends in the two datasets are similar (Fig. 2). Nevertheless, the HYDE and FAO trends are inconsistent with our reconstructed data and other intermittently reported studies from 1960 to 1990. It is also noteworthy that abnormal cropland augments were found in the time-series data of FAO and
HYDE in the 1980s, the NLRB in 2009, and FAO in 2015 (Fig. 2).

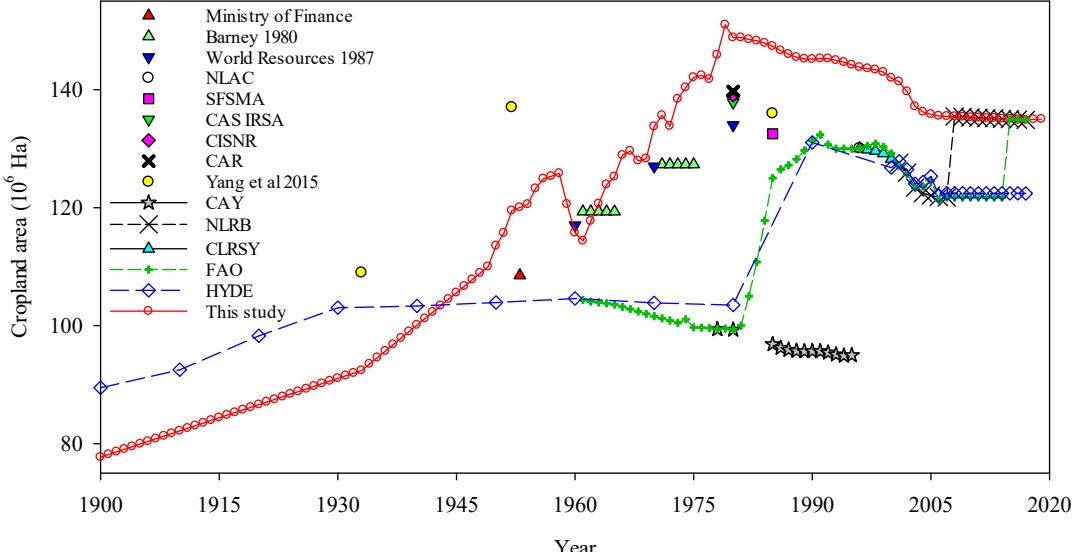

Figure 2. Comparisons of national cropland area data from different sources. (Barney 1980: the number from Barney (1981); World Resources 1987: Institute et al. (1987); NLAC: National Land Administration of China; SFSMA: Soil Fertility Station of the Ministry of Agriculture; CAS IRSA: Institute of Remote Sensing and Digital Earth, Chinese Academy of Sciences; CISNR: Committee of Integrated Survey of Natural Resources; CAR: Committee of Agricultural Regionalization; NLRB: National Land and Resources Bulletin; CLRSY: China Land and Resources Statistical Yearbook)

We compared the provincial cropland areas and coverage percentages using data derived from HYDE, Yang et al. (2015), and our study with officially released NLRB data (Fig. 3). Obviously, the HYDE and Yang et al (2015)'s cropland acreages were moderately underestimated in the high cropland provinces (e.g., Heilongjiang Province) (Fig. 3a; $R^2 = 0.88$, slope = 0.75–0.78), while GlobalCrop30m generally overestimated cropland distribution in most of the provinces (Fig. 3a; $R^2 = 0.87$, slope = 1.1). In comparison, the coverage percentages are also consistent between the datasets (Fig. 3b, c, d). Similar to the cropland acreage, GlobalCrop30m tends to overestimate the cropland percentage in the high cropland coverage provinces (Fig. 3c). Surprisingly, Yang et al.'s (2015) cropland percentage is highly consistent with the percentages obtained in our study, although the two datasets were independently constructed (Fig. 3d; $R^2 = 0.96$, slope = 1.00).

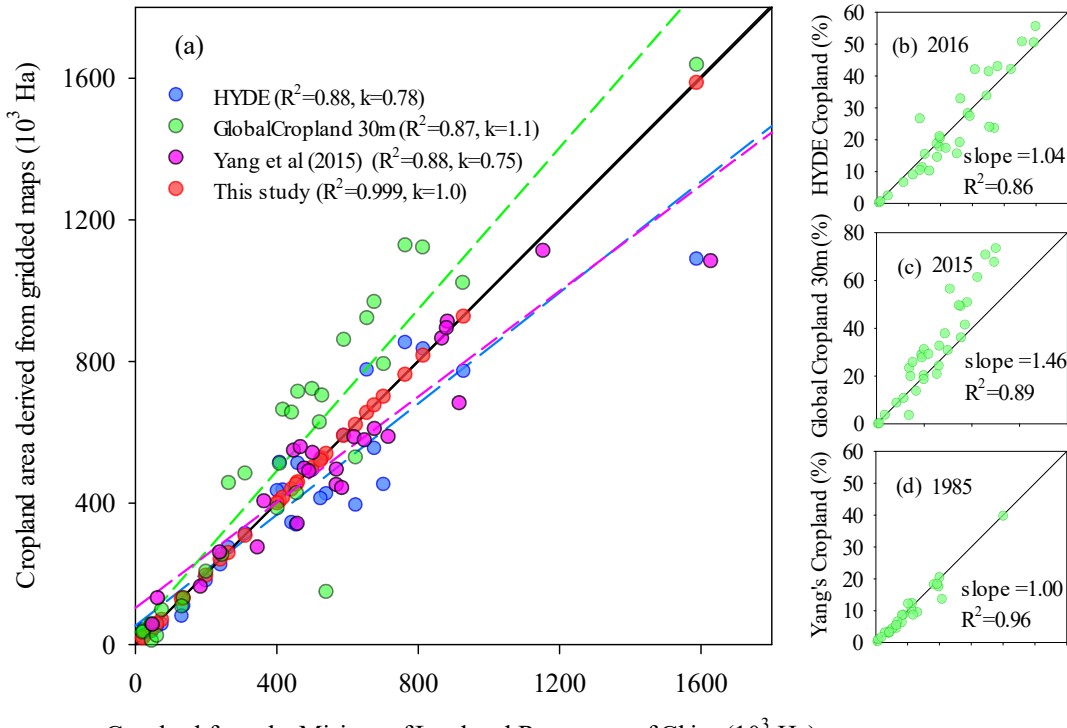

Figure 3. Comparison of cropland (a) acreage and (b) percentage in each province among different gridded maps and inventory data (the black solid line is the 1:1 line and the colored dash lines are the linear regression lines).

We also compared the national cropland acreages from our results and HYDE with other studies on intermittently reported cropland (Table 3). In comparison to the relatively steady cropland of about 103–104 Mha from HYDE during the period of the1930s to the 1980s, the intermittently reported croplands were about 100–144 Mha. Our reconstructed croplands are generally higher than the intermittently reported croplands, except for the period before 1953.

Table 3. Estimates of total cropland area (million hectares) in China from this study, HYDE 3.2, and other studies.

| Data sources | Year | Cropland data from other studies | | This study |
|---|---|---|---|---|
| | | Acreage | HYDE 3.2 | |





| | | | | |
|---|---|---|---|---|
| Yang et al. (2015) | 1933 | 108.98 | 103.06 | 92.40 |
| | 1952 | 144.22 | 103.94 | 119.50 |
| | 1985 | 136.86 | 103.50 | 147.37 |
| Ministry of Finance of the People's Republic of China | 1953 | 108.53 | 103.94 | 120.04 |
| Feng et al. (2005) | 1950 | 100.49 | 103.94 | 113.54 |
| | 1980 | 135.04 | 103.50 | 147.31 |
| Committee of Agricultural Regionalization | 1980s | 139.69 | 103.50 | 147.31 |
| Committee of Integrated Survey of Natural Resources | 1980s | 139.06 | 103.50 | 147.31 |
| Institute of Remote Sensing and Digital Earth, CAS | 1980s | 137.82 | 103.50 | 147.31 |
| Soil Fertility Station of the Ministry of Agriculture | 1985 | 132.52 | 117.27 | 147.37 |
| National Land Administration of China | 1996 | 130.04 | 128.91 | 143.79 |
| Lai et al. (2016) | 2010 | 179 | 122.44 | 135.27 |


### 3.2 Spatial patterns of cropland distribution during different periods in China

We compared the cropland distribution in sets of decades beginning with 1900. Our reconstructed maps showed that the cropland was primarily distributed in three historically cultivated plains in China: the Sichuan Plain, the Northern China Plain, and the Northeast China Plain (Fig. 4). Remarkably, the cropland acreage in the HYDE maps is much lower in the Sichuan

Plain and the Northeast China Plain, while the croplands are much more extensively distributed throughout the rest of China (Fig. 4).

We also compared the cropland change in each of the studied decades by simply calculating the coverage differences. The cropland change patterns were also similar between our study and the HYDE maps, albeit the discrepancies were found in different periods and at different locations. For example, during the period of 1930–1960, the HYDE maps detected large

scale cropland loss between the Yellow River and the Yangzi River. Nevertheless, our data revealed that the region was dominated by intensive cropland expansion during the same period. Moreover, in contrast to the cropland loss found in parts of Northeast China during 1930–1990, our reconstructed maps showed cropland expansion in the corresponding areas (Fig. 4). For the most recent decades, intensive cropland expansions were found in Northwest China and central Northeast China from our reconstructed maps (Fig. 4i). These changes were less often captured in the HYDE maps (Fig. 4r).

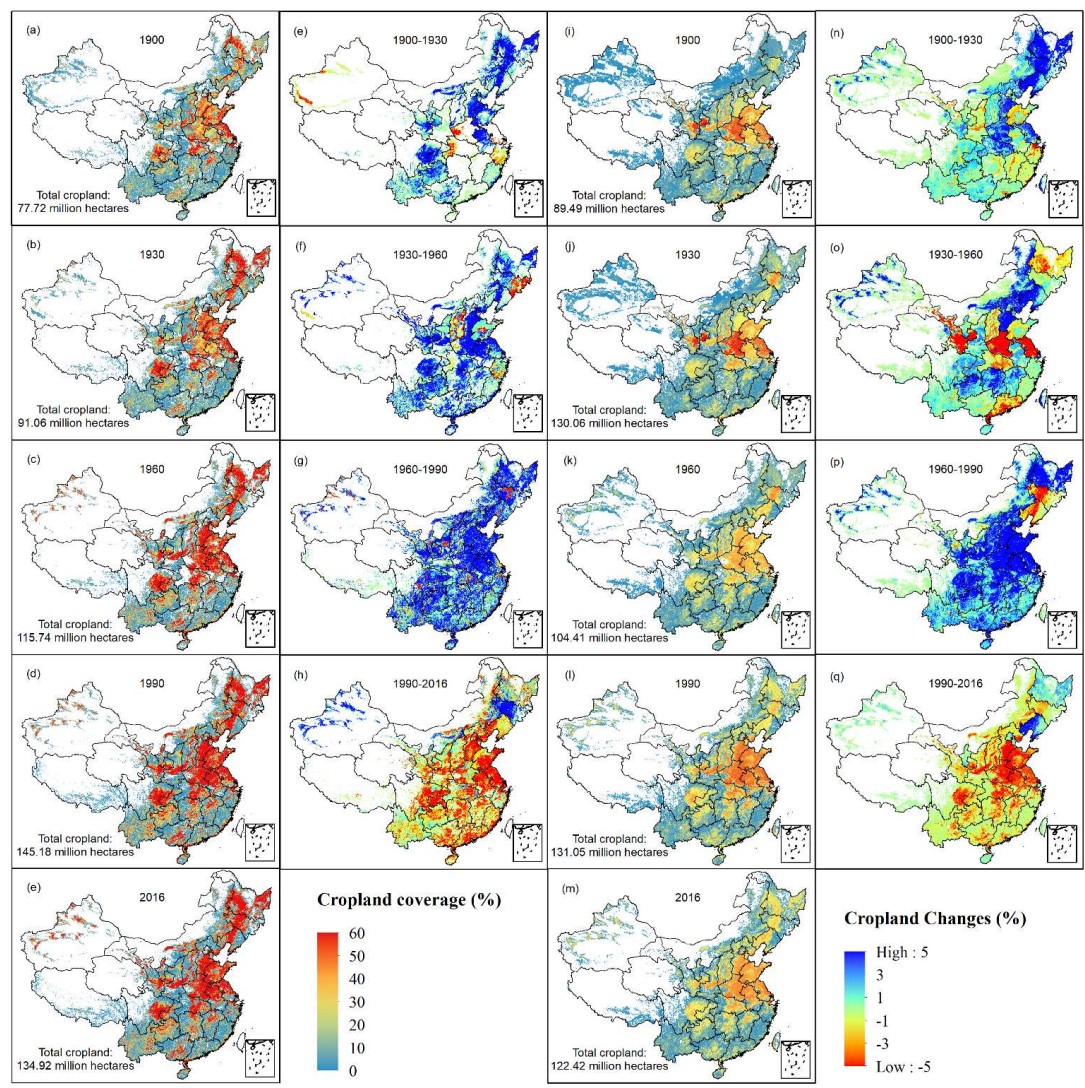


Figure 4. Cropland coverages and changes during different periods in China (left two panels) and HYDE (right two panels) (excluding Taiwan, Hong Kong, Macaw, and the islands in the South China Sea; Fig. 4a-e and Fig. 4i-m indicate 1900, 1930, 1960, 1990, and 2016, respectively).

**3.3 Spatial patterns of cropland distribution during different periods in China**

Limited by the availability of the time-series data, we analyzed and compared cropland abandonment and expansion during the 1900–2016 period, as well as the year of the onset of cropland abandonment and expansion, as derived from our reconstructed maps and the HYDE maps (Fig. 5). In general, the abandonment years mainly occurred during the 1970–2010 period (Fig. 5a, e), while the HYDE-based cropland abandonment year was generally one decade earlier than the year derived

from our maps. Another distinct difference is the distribution of the early cropland abandonment (before 1930), which, in our data, was mainly found in Northeast China (Fig. 6b), in contrast to that found in the Central China, Northeast China, and Northwest China from the HYDE maps (Fig. 6e). In comparison to the abandonment area identified from our maps, the incidence of HYDE-based cropland abandonment was lower in the northern and southwestern regions of China, but it was higher in the northwestern and northeastern regions (Fig. 5b, f, respectively).

315        Conspicuously, a significant difference in the expansion years was observed between the results derived from the two datasets (Fig. 5c, g, respectively). More specifically, in our maps, cropland expansions mainly occurred in the years before 1970, while that it generally occurred after 1970 in the maps derived from HYDE. Furthermore, cropland expansion was much smaller in the central and southern regions of China in the HYDE maps than in our maps (Fig. 5d, h, respectively).

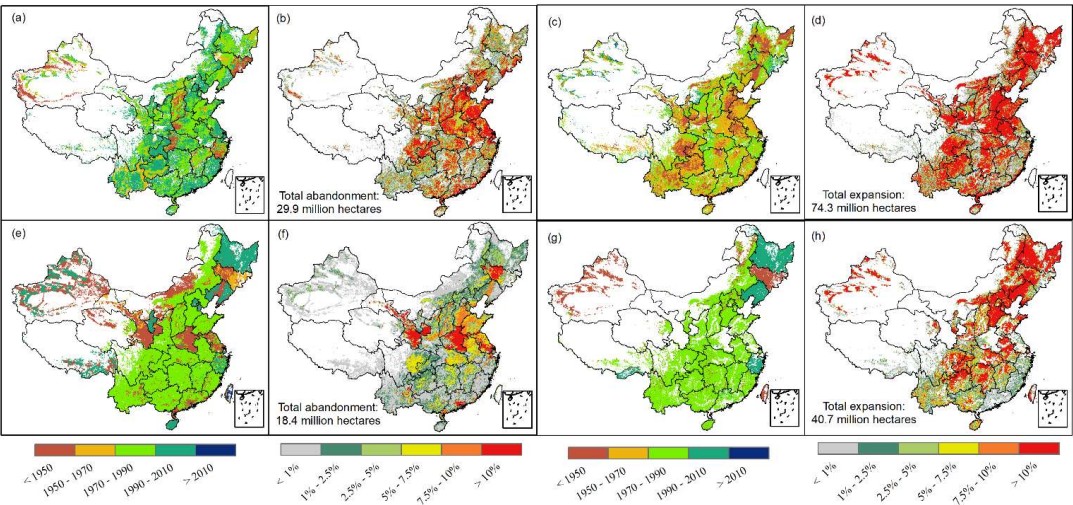

Figure 5. Distribution of cropland (a, e) abandonment and (c, g) expansion year, and (b, f) abandonment and (d, h) expansion percentage in China from 1900 to 2016 (upper panel: this study; lower panel: HYDE).



**4 Data Availability**

All cropland data reconstructed in this study are publicly available via https://doi.org/10.6084/m9.figshare.13356680 (Yu et al., 2020).

**5 Discussion and conclusion**

**5.1 Temporal changes in the national cropland acreage**

Uncertainty about the historical cropland acreage in China is enormous due to the contradictory data reported by official agencies. Many studies have tried to reconstruct historical cropland distribution maps in China, but only a few intermittent years were available for several key years. What is lacking is a continuous cropland dataset with acreage that is consistent with
the data that have been officially released. Since 1949, collections of cropland acreage data were increasingly organized to serve the development of economic plans. The first official estimation of national cropland was 108.53 Mha conducted in 1953 by the Ministry of Finance of the People's Republic of China. However, the methods used in the land survey have several drawbacks (Zheng, 1991). First, the units used in the land survey differed between locations. The traditional metric units used in China were not unified and they referred to different lengths in the 1950s (although all were called "Mu"), resulting in the
significant discrepancy in information about the croplands been surveyed. Second, the underestimation from adopting the "production-to-acreage" extrapolation approach was limited by technology and accessibility in the field survey. Historically, this traditional method has been widely used in China for taxes purpose. Specifically, the approach converts low productivity croplands (e.g., steep lands, marginal lands) into less acreage (~1/3–1/8) being surveying by referring to a "standard productivity" cropland (Perkins, 2017). Thus, it is highly possible that the converted acreage being reported to the local
government greatly underestimated the actual cropland size. Despite these limitations, this method was a benchmark used for updating cropland areas for a few decades after 1949.

The second national cropland estimation (130.04 Mha), the FNLS, was conducted during 1985–1996. This 10-year land survey is the first comprehensive and systematic survey performed in China using many aerial photos and field measurements. Nevertheless, the uncertainties are still very large as deduced from time-unmatched information between the
photos and the field surveys, the low quality of the photos, and manual digitalization. Due to the awareness of the limitations in the estimation, the SNLS was conducted from 2007 to 2009, aiming at delineating each parcel of land being used in China. The SNLS is the most recent, comprehensive survey officially conducted using integrated state-of-the-art technologies, including airborne and spaceborne remote sensing, a geographic information system, a satellite navigation system, etc.

Based on the benchmark data provided, we found that our reconstructed cropland was about 14 Mha (11%) higher
than the area derived from FAO and HYDE for the period of 1990 to 2014. It should be noted that FAO updated the cropland acreage using official data since 2015, causing an abrupt change in the amount of cropland (Fig. 2). Surprisingly, the amount of FAO and HYDE cropland abnormally increased by 28–32 Mha from 1980 to 1990, which contradicted the 4 Mha decline



in cropland acreage revealed in our reconstructed data. This may because the official data reported to FAO was from the CAY in which cropland underestimations have now been officially acknowledged. For the period after 1980, FAO abandoned the

CAY data and used the CLRSY data instead, resulting in an abrupt change in the amount of cropland acreage. A similar abrupt increase in the cropland in HYDE was inherited from FAO, which was used as the basis for the HYDE reconstruction.

In addition to the abrupt changes in the increase in cropland, we found that the FAO data on changes in cropland during the period of 1960 to 1980 were also distorted. Noticeably, in FAO, a persistent decrease in the cropland acreage was identified from 1960 to 1980, which is different from the rapid cropland expansion identified in our reconstructed data (Fig.

2). We found that the distorted trend in the FAO cropland data was rooted in the CAY data that were used in which the decrease in the amount of cropland during the 1960–1980 period has been questioned by many studies (Feng et al., 2005; Zheng, 1991). For example, to encourage cropland expansion, an incentive policy was implemented to allow the newly cultivated lands to be free from taxation and excluded from reporting to the government for the first 3–5 years during that period (Bi and Zheng, 2000; Zheng, 1991). In reality, the newly cultivated lands were never reported to the local government (Feng et al., 2005).

Moreover, many newly cultivated lands were cultivated but "concealed" due to political issues, in order to report a higher crop yield to the local government (Zheng, 1991). Therefore, the cropland area officially reported from 1960 to 1980 has been greatly distorted and should be discarded.

An alternative approach is required to reconstruct data on the national cropland acreage for the period of 1960–1980. Innovatively, Feng et al. (2005) found that cropland area and crop productivity were closely related, which were used to rebuild

cropland acreage for the period of 1960–1980. This approach is grounded in the fact that crop production was strictly controlled and allocated by the government during the planned economic period, rendering the crop production figures unlikely to be distorted and thus reliable as a reference. However, it should be noted that agricultural technology and climatic factors also affect crop production. Therefore, the approach should be limited to the period of the samples used, and extrapolating to other periods may cause bias. Similarly, we reconstructed national cropland acreages from the area-production relationship obtained

from the intermittently reported data from multiple sources from 1949 to 1980 ($R^2$=0.92, Fig. S3, Table S1). Then, we extracted the inter-annual changes in the cropland area and applied that to the baseline year of 1980 to reconstruct the cropland acreage from 1949 to 1979.

Consequently, we reconstructed data on historical cropland acreage by retrospectively extending the area assuming that the recent national survey and historically documented cropland changes are reliable. Our results revealed that most of the

other studies have systematically underestimated the amount of cropland area by about 3–13 Mha. More importantly, HYDE and FAO data not only underestimated the cropland coverage, they also distorted the cropland changes. For example, the maximum cropland coverages were underestimated by 40–50% in two of the historically cultivated plains in HYDE (i.e., the Sichuan Plain and the Northeast China Plain) (Fig. 4, Fig. S4). This is consistent with the results reported in our previous study, which found that HYDE underestimated cropland density in the intensively cultivated area of the US (Yu and Lu, 2018).

Using intensive comparisons of the distribution pattern and change trends with HYDE and FAO data, we are more confident in the reliability of our reconstructed maps because 1) the datasets used in cropland reconstruction are more reliable;

and 2) the cropland distribution is more consistent with the results reported in other studies conducted in China. As previously mentioned, the HYDE maps are impacted by the distorted signals from the FAO data, while our maps were reconstructed from corrected tabular data. Our cropland maps also intensively assimilated the signals from satellite images. For example, the

GFSAD30m product that we used is a high-resolution gridded map derived from Landsat images with assists from over 100,000 reference data samples and other auxiliary data sources (Xiong et al., 2017; Zhong et al., 2017). Moreover, our cropland distribution is more consistent with the findings in previous studies (Li et al., 2010, 2016), which reported that HYDE underestimated the cropland area in Northeast China in the 20th century (Li et al., 2010) and most of the central and northern regions of China in 2000 (Li et al., 2010). Thus, we conclude that our reconstructed maps are more reliable than the HYDE

maps in depicting cropland distribution, while the HYDE maps greatly underestimate the crop density in high cropland coverage regions (Fig. S4). As revealed in our previous study, HYDE allocated cropland to each grid cell according to the weighed maps generated from information from both social and natural indicators, such as urbanization level, population, soil suitability, climate, and topography (Goldewijk et al., 2017; Yu and Lu, 2018). Therefore, HYDE maps can be used as agricultural potential and crop suitability maps, especially in the early period when satellite images were not available (Yu and

Lu, 2018). Nevertheless, it should be noted that the underestimated cropland coverages and the biased change trends require caution when used in biogeochemical simulations.

Overall, our study produced spatially-explicit time-series cropland percentage maps that are consistent with officially released data while assimilating spatial pattern and trend information from various satellite products. Through intensive comparisons, we are confident that our data provide relatively long-term (1900–2016), moderate resolution (5 km × 5 km),

and reliable (national/provincial acreages and interannual variations) sources of cropland maps.

## 5.2 Cropland abandonment and expansion

In this study, we simply analyzed changes in the cropland coverage for sets of decades beginning with 1900 using the first and last cropland maps of each period (Fig. 4). Both the HYDE dataset and our dataset revealed that cropland expansion was the dominant trend of land conversion for the period of 1900–1990, while cropland abandonment nationwide has been significant

for the last period of 1990–2016. The conspicuous cropland loss during this period was also reported in previous studies, which had been related to climate changes and socio-economic factors (Lai et al., 2016; Liu et al., 2005; Ning et al., 2018). Consistent with the findings reported by Liu et al. (2014) and Lai et al. (2016), we also found that the cropland area decreased in the southern region of China but increased in the northern region for the last period of 1990–2016. However, our cropland shrinkage (-9.91 Mha) from 1990 to 2010 is different from the minor cropland increase (1.48–1.82 Mha) from the late 1980s

to 2010 reported in Liu et al. (2014) and Lai et al. (2016). This may be attributed to the different cropland classification systems adopted in the two datasets. Our data only included traditional croplands, such as lands cultivated and temporally fallowed. In comparison, the data in Liu et al. (2014) and Lai et al. (2016) refer to arable land, which includes traditional cropland as well as tideland and gardens for agricultural production (e.g., fruits, vegetables, mulberries). Thus, the two studies combined suggest a decrease in traditional major crops (e.g., grains) and increases in other types of agricultural crops during the period. This is



consistent with Liu et al.'s (2018) findings that documented significant reductions in grain crops based on data collected from 2341 counties in China. Surprisingly, the number of counties in which fruits, vegetables, and edible oil crops were cultivated increased from <1% in 1990 to ~5% in 2010 (Liu et al., 2018). This change is understandable as rapid economic development drove demands for more diversified agricultural products.

To further examine cropland expansion and abandonment during the entire period of 1900 to 2016, cropland
conversions were identified using two approaches specific for each of the two cropland datasets. For our reconstructed cropland, the expansion occurred in the grid cells that were converted from noncropland in 1900 to cropland in 2016, while abandonments are seen in the grid cells that were historically cultivated but were converted to noncropland in 2016. The abandonment and expansion gird cells were aggregated from 100 m Boolean type to 5 km percentage maps. In comparison, since cropland areas were described as a coverage fraction in the HYDE dataset, cropland conversions were directly extracted
by comparing the difference between the maximum coverage and the coverage in 2016 (abandonment), and the difference between 1900 and 2016 (expansion). In total, we found the abandoned and expanded cropland areas were 29.9 Mha and 74.3 Mha in China from 1900 to 2016, respectively; in the HYDE dataset, it was 18.4 Mha and 40.7 Mha, respectively, for the same time period. In general, HYDE underestimated cropland abandonment and expansion by over 11 Mha and 33 Mha.

Both datasets showed that abandoned croplands were centered at the northern region of China, although the magnitude
differed by locations (Fig. 5b, f). The largest discrepancy is the most intensive cropland abandonment derived from the two datasets. The high cropland abandonment found from the HYDE dataset was concentrated in the northwestern (Shaanxi Province), central (Henan Province), and northeastern (Jilin Province) regions of China, which were not detected in our maps (Fig. 5b, f). In comparison, the high cropland abandonment found from our maps in the southwestern and northern regions of China were missing in the HYDE dataset. The difference may be related to the approach used. The cropland abandonment
year indicates the most recent year that the conversion occurred from our reconstructed maps, while the HYDE-abandonment year represents the date that the cropland coverage began to decrease.

Since implementation of the Grain for Green Project in 1999, a total of 26.62 Mha of cropland was converted back to woodland by 2016; 86% of this occurred in the 1990s and the 2000s (National Forestry and Grassland Administration of China, 2018). This is also visible from the rapid cropland decline in our reconstructed time-series data (Fig. 2), and consistent
with the large scale cropland abandonment after 1990 (Fig. 4). Ning et al. (2018) reported that ~2.04 Mha of cropland was converted to other types of land use and ~1.55 Mha of other land was converted to cropland, resulting in a net loss of 0.49 Mha (0.36%) of cropland area from 2010 to 2015. This is similar to our study's finding; based on statistics from NLRB we found that the conversions between cropland and noncropland were 1.85 Mha and 1.58 Mha, respectively, leading to a net loss of 0.27 Mha (0.2%) during the period.

Cropland expansion timing and magnitude significantly differed between the two datasets used (Fig. 5c, g). Our dataset revealed a more evident cropland expansion in the southern and central regions of China. Moreover, the timing of the cropland expansion pattern is vastly different; our map showed a much earlier cropland reclamation over the entire study area, except for the northwestern and the central northeastern regions of China. This great discrepancy can be primarily attributed



to the methods that were implemented. Our 5-km expansion percentage map was resampled using a grid-by-grid comparison

of the 100-m Boolean type data. For HYDE, it is impossible to directly time the expansion from the 5 arc-min (~8 km) cropland

percentage map. Alternatively, in this study, the occurrence of the largest cropland expansion was used to represent the rough

date of expansion in HYDE. Thus, the dominant HYDE-based cropland expansion that was detected occurred in the 1970–

1990 period, which is consistent with the abrupt change in the increase in the amount of cropland (Fig. 2), but is a misleading

signal as we explained. Instead, the expansion-year map produced from our data indicates the latest year of conversion from

noncropland to cropland, which primary occurred before 1970 (Fig. 5c). Our data showed that the most significant cropland

expansion since 1990 was found in the northwestern region of China (i.e., Xinjiang Province), which is the reclaimed cropland

in the dry region assisted by the development of oasis agriculture (Han, 2009; Ning et al., 2018). Despite the noticeable

cropland abandonment in the northwestern region (Fig. 5a), a more intensive reclamation resulted in a substantial net increase

of cropland in that area. It is important to note that the other hot spot of recent cropland expansion is located in the Northeast

China Plain, which is also documented in other studies (Xia et al., 2016; Zhang et al., 2018). The intensive cropland

reclaimation might have triggered a significant amount of carbon emissions as the region is well-known for its high carbon

storage; our updated cropland maps are expected to be helpful for such an assessment.

    Specifically, we found that the cropland abandonment and expansion in the mid-eastern region of Inner Mongolia

are consistent with the results reported in Dong et al. (2011). The region experienced rapid land cover conversions due to

conflicts between the increasing demands for food and the pressure for environmental protection. Although a net of 1.2 Mha

of cropland was reclaimed in the region from 1990 to 2005, the land conversions shifted from being expansion-dominated to

being abandonment-dominated after 2000 (Dong et al., 2011). This might be attributed to the implementation of the Grain for

Green policy. Accordingly, our abandonment and expansion maps showed that new cultivations occurred in the region in the

1990s. Thus, government policy incentives, improvements in farming techniques, and increases in the population are the three

major drivers of recent cropland expansion in China. For example, the Chinese government sponsored improvement of

irrigation projects and agricultural mechanization, resulting in subsequent agricultural development in the provinces in central

China since the 2000s, which can be seen in the maps reconstructed in this study (Fig. 5c).

    Although a large degree of uncertainty remains in HYDE land use maps, they have been widely used by the global

change modeling community (e.g., Van Oost et al., 2007; Tian et al., 2013; Wang et al., 2020). However, the HYDE cropland

maps might cause significant bias if they are used in regional and local biogeochemical simulations. Based on the HYDE maps,

Wang et al. (2020) and Zhang et al. (2020) estimated $N_2O$ emissions globally and in China's croplands from 1961 and 1949,

respectively. However, we expected that the $N_2O$ emission results might have been biased as fertilizer was applied in a smaller

cropland area (e.g., 103.99 Mha in HYDE vs. 132.68 Mha in our data during 1960–1980). Moreover, although the

underestimation of cropland was only 10% during the recent period of 1990–2016 (126.34 Mha in HYDE vs. 140.60 Mha in

our data), the use of nitrogen as a fertilizer has increased by more than three-fold during this period (Gu et al., 2017). Thus,

the nutrient cycles and water pollution levels need to be reassessed using updated cropland maps. Therefore, the improved



cropland maps we reconstructed in this study are expected to help reduce the bias and obtain more accurate biogeochemical simulations.

**Author contributions**

ZY designed the work. XJ and XY provided cropland maps in 1933, 1952, and 1985 for reconstruction of the dataset. ZY, XJ, XY, and LM reviewed and edited the writing. ZY prepared the manuscript and wrote the final paper with contributions from all the coauthors.

**Competing interests**

The authors declare that they have no conflict of interest.

**Acknowledgement**

We greatly acknowledge data entry assistance from Yanli Dong and Jing Guo. The CNLUCC data set is provided by Data Center for Resources and Environmental Sciences, Chinese Academy of Sciences (RESDC) (http://www.resdc.cn). This work was supported by National Natural Science Foundation of China (No. 32001166), Jiangsu Key Laboratory of Agricultural Meteorology Foundation (JKLAM2004), and the Startup Foundation for Introducing Talent of NUIST (No. 2019r059).

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
