# Peer review of "A historical reconstruction of cropland in China from 1900 to 2016"

_Earth System Science Data, 2020_

## Author Comment (AC2)

Spatial mapping of croplands distribution is essentially needed for multiple agricultural and environmental studies. In particular, the long-term croplands spatial datasets, like the one introduced by this study, enable the long-term tracking of cropland changes and analyzing their driving factors. Hence, I believe that the current study is introducing a very useful cropland dataset and so, it has great potential to be published in ESSD after considering a few minor revisions. I have listed three comments below for your consideration.

Response: We thank the reviewer for valuing our work!

- What is the final spatial resolution of the produced cropland grids? It will be better to have this information in the abstract.

  Response: We thank the reviewer for the suggestion. The final spatial resolution of the cropland product is 5 km $\times$ 5 km. We will add this information to the abstract.

- It is clear that the cropland areas at the provincial level estimated from the constructed cropland dataset were compared with other earlier studies, and all were compared to official statistical data. However, I believe that this comparison might be biased because the official statistical data (from NLRB) was part of the tabular data used to construct the final cropland layers, right? That is why the correlation between the produced cropland layer and NLRB data was very high (R-Square = 0.999, k=1.0, in Figure 3). Therefore, I believe the accuracy assessment, through traditional error matrix and estimated OA, PA, Kappa, etc., can be performed here for selected years when crowdsourcing validation samples (or field samples) are available to reflect the efficiency of the cropland spatial allocation method introduced by the current study. Some crowdsource validation sets are publicly available and can be used for validation such as the one from Geo-Wiki (https://www.nature.com/articles/sdata2017136) and Global validation sample set v1 (http://data.ess.tsinghua.edu.cn/).

  Response: We thank the reviewer for the suggestion. Yes, we used the official statistical data from the National Land and Resources Bulletin (NLRB) for the comparisons of different cropland datasets. The NLRB is also used in the reconstruction of the cropland in this study since it is the most authoritative report officially released by the Chinese government. We provided the provincial level comparison to illustrate the model performance in the reconstruction of the data.

  The reviewer's suggestion is very helpful because the ground-true observations provide the validation at the finer resolution. However, directly validation is not suitable for the cropland product reconstructed in this study as traditional error matrix and accuracy assessment is designed for validation of classified land cover maps. The cropland product, as well as the HYDE data, describes cropland in percentage, and thus the site-based validation is not directly applicable.

Despite such limitation, we agree that the validation is important and necessary. Therefore, we chose the intermediate product (Boolean type map at 100-m resolution before aggregating to 5-km cropland percentage map) derived from multiple data sources for the validation using the Global validation sample set v1 (http://data.ess.tsinghua.edu.cn/). Note that the intermediate product describes the high possibility of absence (0) or presence (1) of cropland in each grid cell, but other land cover types were not available. Therefore, we directly compared the intermediate product with cropland sites provided in the Global validation sample set. We found that, in total of the 356 cropland sites from mainland China, 219 (62%) of the sites were correctly identified in our intermediate product (from 2001 to 2010), which is in the range of 198 (56%) to 248 (69%) identified from the MODIS product (MOD12Q1, from 2001 to 2014) and the GFSAD30 (2015), respectively. However, since the intermediate product is not the focus of this study, we will add this information to the supplementary file.

- In figure 1, GFSAD30 was produced for 2015 and so I think it belongs to the 2010s rather than the 2000s, right?.

  Response: We thank the reviewer for pointing out this. Exactly, the GFSAD30 was produced for 2015 and it was mistakenly placed in Figure 1. The figure will be updated to:

---

## Author Comment (AC3)

A spatially-explicit cropland distribution time-series dataset is necessary for the accurate assessment of biogeochemical processes in terrestrial ecosystems and their feedback to the climate system. This study reconstructed a continuously covered cropland distribution dataset in China spanning from 1900 to 2016 by assimilating multiple data sources and identified the abandonment and expansion of cropland, which has important contribution to this research area. However, some questions are as follows:

Response: We thank the reviewer for valuing our work!

1. Many scholars (Li, Yang, Wei et al.) have done the research on the gridding allocation of cropland in China during the 300 year. Why do you choose 1900 as the starting time point to repeat the allocation? Whether you have more dependable historical data sources for 1900 to 1949 or have you revised some time section's value?

   Response: We thank the reviewer for pointing out this. Exactly, many scholars have done great jobs in rebuilding the cropland distributions in China for the past 300 years. We chose 1900 as the starting time point because of two reasons. First, previous researches have done very good jobs in allocating cropland spatially in early periods (e.g. He et al 2003, Wei et al 2019, Ye et al 2009). The approaches used in these previous studies have more complex mechanisms than our model, while our model highly relies on the abundance of gridded data available. Since we do not have better data sources in early period, our major focus is to correct the biases in cropland area changes for the period since 1949. Second, we simply extrapolated cropland distributions from 1949 back to 1900 for serving the simulations demands of the modeling community, which generally requires a historical, annual cropland maps for model input. According to the previous studies (e.g. Li et al 2016, Yang et al 2015), the cropland changes from the end of the 19[th] century to the early 20[th] century are relatively minor. Therefore, we simply used the trends of cropland changes during the periods of 1887-1933 and 1933-1952 to gap-fill the data during the period of 1900-1948 in this study. The cropland data in 1887, 1933, and 1952 were obtained from Yang et al (2015).

   References:

   He, F. N., Tian, Y. Y., & Ge, Q. S. (2003). Spatial-temporal characteristics of land reclamation in Guanzhong region in the Qing Dynasty. Geographical Research, 22(6), 687-697.

   Li, S., He, F., & Zhang, X. (2016). A spatially explicit reconstruction of cropland cover in China from 1661 to 1996. Regional Environmental Change, 16(2), 417-428.

Wei, X., Ye, Y., Zhang, Q., Li, B., & Wei, Z. (2019). Reconstruction of cropland change in North China Plain Area over the past 300 years. Global and Planetary Change, 176, 60-70.

Yang, X., Jin, X., Guo, B., Long, Y., & Zhou, Y. (2015). Research on reconstructing spatial distribution of historical cropland over 300 years in traditional cultivated regions of China. Global and Planetary Change, 128, 90-102.

Ye, Y., Fang, X., Ren, Y., Zhang, X., & Chen, L. (2009). Cropland cover change in Northeast China during the past 300 years. Science in China Series D: Earth Sciences, 52(8), 1172-1182.

2. Whether this study considered the difference between pure and mixed cropland grid cells, or different proportion range of cropland grid cells in different remote-sensing products when they are used for reconstruction?

Response: We thank the reviewer for pointing out this. For some of the datasets, the grids are classified to pure and mixed croplands. A typical example is the MODIS land cover product under the IGBP classification system, which includes land cover type categories of croplands and cropland/natural vegetation mosaics. However, the classifications of the datasets used in this study do not include mixed cropland type, although the grids definitely include mixed cropland in different datasets. Specifically, we chose Plant Functional Types (PFT) classification for MODIS data, in which cropland is divided into cereal cropland and broadleaf cropland (see explanation below). Therefore, the grids are either cropland or non-cropland in the datasets used in this study. For example, the GlobeLand30 and GFSAD30m are Boolean type data specifically designed for describing cropland distribution. Thus, we avoided using "mixed cropland" classification type. Croplands were separated from other land cover types and converted into Boolean type for model input.

Note that all these datasets, including high-resolution GlobeLand30 and GFSAD30m, have both pure and mixed cropland grid cells. However, we were unable to separate pure and mixed cropland grid cells from these datasets. Therefore, we designed the model to give priority to higher resolution gridded product in cropland allocation. For example, the MODIS data with 500-m resolution has a higher priority than the contemporary, 1-km resolution data (e.g. GLC2000), but a lower priority than the 30-m resolution data (e.g. CNLUCC). We chose the PFT classification system for MODIS land cover product, in which cropland are defined as grid cells dominated by cultivated crops (>60%). Therefore, all the datasets were converted to Boolean-type images, namely each grid cell is either cropland or other non-cropland, and the cropland grids within each dataset were treated equally. By giving different priorities to each dataset,

the weighting scores help reduce the possibility to allocate cropland to grids with low crop coverage.

3. For satellite-data period and pre-satellite era (1900-1979), this paper has used different spatializing approach. For gridding images, cropland fraction, distance to urban, correlation with previous years and resolution were considered for the weighted value in the potential cropland map in this paper. So, the results, to a large extent, will be decided by the dependability of different satellite products. Whether you have think about the priority of high resolution and dependability of remote-sensing products ¼Ÿ Or change another word, for China, maybe GlobalLand30 or Gong's data is more fit for?

Response: This is a great suggestion! We agree with the reviewer's suggestion and included GlobalLand30 or Gong's data (FROM-GLC) for updating the reconstruction of the cropland during the period before 1979. More specifically, the two datasets were given the same priority as the Yang's data in cropland reconstruction. We believe that Yang's data provides important cropland distribution signals in the early period, and GlobalLand30 and Gong's data (FROM-GLC) also contain the footprint of the early cropland information. The cropland maps have been reconstructed and undated (see https://figshare.com/articles/dataset/ChinaCropland_zip/13356680).

4. Reconstruction of low cultivation ratio regions maybe has large uncertainty, especially Xinjiang and XiZang, where the results of crop distribution area or intensity all exists unreasonable. You had better analyze it more from the view of method uncertainty or discrepancy of different sets of products.

Response: We thank the reviewer for the suggestion. We rechecked and compared the cropland distribution in low cultivation region of Xinjiang and Xizang (Tibet). Indeed, the cropland distributions in these two provinces are quite different between studies. Generally, our reconstructed cropland was similar to these published results or products in Xinjiang (e.g. cropland product from: http://www.dsac.cn/DataProduct/Detail/20081129, Liu et al (2005)), while the discrepancy in Xizang is much larger (e.g. comparing with Li et al (2016), Liu et al (2005), and the product from: http://www.dsac.cn/DataProduct/Detail/20081127). We found that our reconstructed cropland in Xizang in 2016 (4446 km$^2$) is the same as the official released data from the Ministry of Land and Resources of China, which is also close to the cropland areas released in the local government report http://nynct.xizang.gov.cn/xwzx/bmdt/202007/t20200715_162472.html) and Liu et al (2005) (4422 to 4690 km$^2$ in the period of 2000 to 2019). Thus, the major discrepancy lies in the spatial distribution of the croplands.

Therefore, we summarized the cropland area in Xinjiang and Xizang from

the top four priority datasets used in our model (Table 1). During the 2010s, the reconstructed cropland maps were mainly determined by the top four datasets at 30-m resolution, namely the GFSAD30m, CNLUCC, GlobeLand30m, and FROM-GLC. We found that FROM-GLC has a much broader distribution of low cultivation grids (Figure 1b), which contributed to the wide coverage of low cropped intensity area (Figure 2a), especially in Xizang region. Therefore, we reduced the priority of the FROM-GLC dataset (priority ranked after the other three 30-m datasets), and reconstructed the cropland in 2010s (Figure 2b). We found that this adjustment reduces the low cultivation area, especially in Xizang region. We believe the improvement help decrease the uncertainty in spatial distribution of the reconstructed cropland maps. The updated croplands have been uploaded and replaced the old version online (see https://figshare.com/articles/dataset/ChinaCropland_zip/13356680).

Table 1. Summary of cropland area in Xizang and Xinjiang from different datasets

| Region | Datasets | Cropland area (km$^2$) | Year |
|---|---|---|---|
| Xizang (Tibet) | GFSAD30m | 1213.99 | 1990-2017 |
| | FROM-GLC | 2947.54 | 2017 |
| | GlobeLand30m | 5080.42 | 2010 |
| | CNLUCC | 7672.57 | 2018 |
| Xinjiang | GFSAD30m | 62925.14 | 1990-2017 |
| | FROM-GLC | 83905.41 | 2017 |
| | GlobeLand30m | 82188.25 | 2010 |
| | CNLUCC | 90290.61 | 2018 |

[Figure]

Figure 1. Comparison of cropland distributions from (a) GFSAD30m, (b) FROM-GLC, (c) GlobeLand30m, and (c) the CNLUCC (cropland maps were directly resampled to 5-km; the low cultivation area (cropland coverage <0.5% and 0.5%-1.0%) was highlighted in red and yellow).

[Figure]

Figure 2. Comparison of the reconstructed cropland in 2016 before and after adjustment (a: before adjustment; b: after adjustment)

Reference:

1) Li, S., He, F., & Zhang, X. (2016). A spatially explicit reconstruction of cropland cover in China from 1661 to 1996. *Regional Environmental Change*, *16*(2), 417-428.
2) Liu, J., Liu, M., Tian, H., Zhuang, D., Zhang, Z., Zhang, W., ... & Deng, X. (2005). Spatial and temporal patterns of China's cropland during 1990–2000: an analysis based on Landsat TM data. Remote sensing of Environment, 98(4), 442-456.
3) Cropland report of Xizang: http://nynct.xizang.gov.cn/xwzx/bmdt/202007/t20200715_162472.html (last accessed April 18th, 2021)

5. For data in 1900-1979, what's the difference between Yang's constrained CA models with other scholars' method? For example, Wei's "the partition and layering-based gridded method" or Li's method are all based on land suitability for cultivation affected by climate, soil and elevation etc. What's the significance of this study on methodology?

Response: We thank the reviewer for pointing out this. Many researchers have reconstructed historical land use using "top–down" decision-making method to match overall cropland area to land parcels using land arability and universal parameters. In comparison, Yang's constrained Cellular Automaton (CA) model is a "bottom–up" model considering the concentrated distribution of cultivated land and various factors influencing cropland distribution, including environmental and human factors. In Yang et al (2015), the CA model takes a historical cropland area as an external variable and the cropland distribution in 1980 as the maximum potential scope of historical cropland. In reconstruction of historical cropland distribution, Yang et al (2015) selected elevation, slope, water availability, average annual precipitation, and distance to the nearest rural settlement as the main influencing factors of land use suitability. Besides, an available labor force index is used as a proxy for the amount of cropland to inspect and calibrate these spatial patterns.

The figure 3 illustrated the differences of using "top–down" and "bottom–up" models in cropland reconstructions. The "top–down" model in combined with the proportional allocation approach tends to produce a historical cropland map with spatial pattern highly close to the recent cropland map (Figure 3a). Examples are the studies such as Li et al (2016) and Wei et al (2016). In comparison, the "top–down" model in combined with the Boolean allocation approach produces a cropland map less similar to the recent cropland (Figure 3b), in which the lower suitability grids were prioritized to be removed in cropland reconstruction (e.g. HYDE 1.1 data). While Yang's "bottom-up" method tends to remove the scattered, lower suitability grids in cropland reconstruction, which is also adopted in Long et al (2014). Therefore, Yang 's model assumes that scattered, fragmented cropland girds were cropped later and should be removed with priority in historical cropland reconstruction.

[Figure]

Figure 3. Comparisons of spatial patterns cropland reconstructed from different approaches

---

## Author Comment (AC4)

Reconstructing of long-term statistical areas as well as the spatial distribution of historical cropland are essential to track the dynanmics of the agriculture development and to analysis the driving factors from natural and anthropological aspects. The manuscript collected and integrated different data source to reconstruct time series cropland layers since 1900. The topic is of interests to researchers and has potential to be published on ESSD. However, some major issues must be addressed before considered in ESSD.

Response: We thank the reviewer for valuing our work!

1.When working on the spatializing of cropland distribution, the first type of gridded images was used assuming higher weighting factors with a higher value given to a higher resolution image. However, did the authors test the consistency or inconsistency of different existing land cover dataset? At many cases, the consistency at many regions is low which introduced high uncertainty at the cropland allocation steps.

Response: We thank the reviewer for the suggestion! Exactly, the consistency in many regions is low for different gridded products. We did briefly and visually check the consistencies between different products, and there are few general patterns. First, the consistency is generally higher between high-resolution products than the consistency between high and low-resolution products. Second, the consistency is generally higher in more intensively cultivated regions. As the reviewer pointed out, the low consistency regions are therefore high in uncertainty, which also majorly distributed in the low cropland coverage area. For example, we compared the four 30-m resolution products used in reconstructing cropland in 2010s (i.e. the GFSAD30m, CNLUCC, GlobeLand30m, and FROM-GLC), and found that the major differences were located in the low cultivation area (Figure 1). Therefore, the gridded products are differed in satellite sensors, noise sources, spatial/temporal/spectral resolutions, and algorithms used, which make it difficult to directly compare these products. Fortunately, they were all validated before released for public use, indicating they are capable of capturing cropland distribution/change signals to a certain extent.

[Figure]

Figure 1. Comparison of cropland distributions from (a) GFSAD30m, (b) FROM-GLC, (c) GlobeLand30m, and (c) the CNLUCC (cropland maps were directly resampled to 5-km; the low cultivation area (cropland coverage <0.5% and 0.5%-1.0%) was highlighted in red and yellow).

The high-resolution datasets were more intensively validated using ground-observed data. For example, over 38, 000 samples used in validations of FORM-GLC products (Gong et al 2013, Yu et al 2013). Based on the preliminary check and to reduce the uncertainty, we therefore develop the reconstruction model based on two assumptions. First, higher priorities were given to higher resolution gridded products. Second, higher priorities were given to grid cell more frequently identified as cropland using multiple datasets. This approach used in our model is targeted at reducing the uncertainty introduced by a single product. Besides, according to the suggestion raised by the other reviewer, we also validated the accuracy of the intermediate product (Boolean type map at 100-m resolution before aggregating to 5-km cropland percentage map) using the Global validation sample set v1 (http://data.ess.tsinghua.edu.cn/) (see more details in the response to the third comment).

References:

1) Gong, P., Wang, J., Yu, L., Zhao, Y., Zhao, Y., Liang, L., ... & Chen, J. (2013). Finer resolution observation and monitoring of global land cover: First mapping

results with Landsat TM and ETM+ data. International Journal of Remote Sensing, 34(7), 2607-2654.

2) Yu, L., Wang, J., & Gong, P. (2013). Improving 30 m global land-cover map FROM-GLC with time series MODIS and auxiliary data sets: a segmentation-based approach. International Journal of Remote Sensing, 34(16), 5851-5867.

2.When spatializing the cropland data from statistical area to grid level, why first generated 100m binary cropland & noncropland map and then resampled to 5km grided percentage map? Is it more straightforward to generate the possibility of cropland (0-100) at 5km grid level?

Response: We thank the reviewer for the suggestion! Indeed, it is more straightforward to understand by generating the possibility of cropland at 5 km grid maps. However, the method used in this study is derived from the model developed in our previous study (Yu and Lu 2018). This is a "top-down" model with a general process to assimilated different sources of datasets. Based on the results and analysis, we found that the updated model has decent performance in cropland reconstruction.

Reference:

Yu, Z. and Lu, C.: Historical cropland expansion and abandonment in the continental U.S. during 1850 to 2016, Glob. Ecol. Biogeogr., 27(3), 322–333, doi:10.1111/geb.12697, 2018.

3.After 1980s when Landsat imageries became available, why not to use multi-annual actual satellite data as a validation source to prove the effectiveness of the spatializing approach?

Response: We thank the reviewer for the suggestion. This is a good method to validate our results and model. However, to use the Landsat imageries, a classification approach will be needed to convert the images into gridded cropland maps or classified land cover maps to compare with our reconstructed maps. Since many of the input data products used in our model were derived from Landsat imageries and have been intensively validated, it seems repetitive to perform the validation using Landsat imageries. For example, FORM-GLC maps were derived from Landsat TM/ETM+ and MODIS data and validated using > 38000 samples (Gong et al 2013, Yu et al 2013); the Globeland30m was derived from Landsat TM/ETM+/OLI, and HJ-1 data and validated using >230,000 samples (Jun et al 2014).

We agree that the reviewer's suggestion is very helpful. Therefore, in combined with the other reviewer's suggestion, we used the ground-true observations to validate our

spatializing approach. Despite that direct validation of our 5 km cropland product is not suitable as traditional error matrix and accuracy assessment is designed for validation of classified land cover maps. The cropland product we reconstructed describes cropland in percentage, and thus the site-based validation is not directly applicable. Therefore, we chose the intermediate product (Boolean type map at 100 m resolution before aggregating to 5 km percentage map) derived from multiple data sources for the validation using the Global validation sample set v1 (http://data.ess.tsinghua.edu.cn/). The intermediate product describes the high possibility of absence (0) or presence (1) of cropland in each gridcell, but other land cover types were not available. Therefore, we directly compared the intermediate product with cropland sites provided in the Global validation sample set. We found that, in total of the 356 cropland sites from mainland China, 219 (62%) of the sites were correctly identified in our intermediate product (from 2001 to 2010), which is in the range of 198 (56%) to 248 (69%) identified from the MODIS product (MOD12Q1, from 2001 to 2014) and the GFSAD30, respectively. However, since the intermediate product is not the focus of this study, we added this information to the supplementary file. This also revealed that our spatializing approach is reasonable.

Reference:

Jun, C., Ban, Y., & Li, S. (2014). Open access to Earth land-cover map. Nature, 514(7523), 434-434.

Some minor comments:

1.CNLUCC products are illustrated in figure 1 at different time period but in the description of the figure 1, only CAS1990 is described. Are those two different sources ¼ŸIn Figure 2, both CNLUCC and CAS1990 are not included as a comparison source. Is the ignorance of those dataset by purpose or by mistake?

Response: We thank the reviewer for pointing out this. It is confusing for readers as CNLUCC and CAS1990 are produced by two research groups from the Institute of Geographic Sciences and Natural Resources Research, Chinese Academy of Sciences. Therefore, we added a citation for CAS1990 and also added the description of CNLUCC for clarification:

"CAS1990: Land Cover data originated from 1:1,000,000 land use map produced by the Institute of Geographic Sciences and Natural Resources Research, Chinese Academy of Sciences (Ran and Li 2006)".

"CNLUCC: China Land Use and Cover Change was provided by the  Data Center for Resources and Environmental Sciences, Chinese Academy of Sciences (RESDC) (http://www/resdc.cn) (Xu et al 2005)."

We thank the reviewer for pointing this out. Yes, the cropland areas of CNLUCC and the other five datasets (i.e. IGBP Data and Information System, UMD Land Cover, GLC2000, CAS1990, and WESTDC Land Cover Product 2.0) were not included in the figure. To increase the readability of the figure, we averaged the cropland area from the five datasets and indicated by Ran's data in the new figure. The new figure is updated to include CNLUCC and Ran's data:

[Figure]

Figure 2. Comparisons of national cropland area data from different sources. (Barney 1980: the number from Barney (1981); World Resources 1987: Institute et al. (1987); NLAC: National Land Administration of China; SFSMA: Soil Fertility Station of the Ministry of Agriculture; CAS IRSA: Institute of Remote Sensing and Digital Earth, Chinese Academy of Sciences; CISNR: Committee of Integrated Survey of Natural Resources; CAR: Committee of Agricultural Regionalization; NLRB: National Land and Resources Bulletin; CLRSY: China Land and Resources Statistical Yearbook; Ran's data:   The average of five datasets prepared by Ran and Li (2006) and Ran (2013), including IGBP Data and Information System, UMD Land Cover, GLC2000, CAS1990, and WESTDC Land Cover Product 2.0; CNLUCC: China Land Use and Cover Change dataset obtained from the Data Center for Resources and Environmental Sciences, Chinese Academy of Sciences.)

References:

1)   Ran, Y. and Li, X.: Comparison report of the four 1-km land cover products of China, Lanzhou., 2006

2) Xu, X. L., Pang, Z. G., & Yu, X. F. (2005). Spatial-Temporal Pattern Analysis of Land Use/Cover Change: Methods &Applications. Science and Technology Literature Press: Beijing, China, 90-130.

2.The authors compared the provincial cropland areas and coverage percentages using data derived from HYDE, Yang et al. (2015), and our study with officially released NLRB data (Fig. 3). Do the scatter points include only one year or all years of NLRB data?

Response: We thank the reviewer for pointing out this. For HYDE data, we compared the cropland from the two datasets for the period of 2000 to 2016. Besides, the Global Cropland 30m were indicated using "GFSAD30m" (in the former version, both "GFSAD30m" and "GlobalCropland30m" were used). The GFSAD30m was compared with the NLRB cropland in 2015. For Yang et al (2015), the cropland area was compared with the reconstructed cropland area in 1985 from NLRB, CLRSY, and CAY data. We also added this information in the figure title for clarification. The figure has been updated to:

[Figure]

Figure 3. Comparison of cropland (a) acreage and (b) percentage in each province among different gridded maps and inventory data (HYDE data was the average cropland area from 2000 to 2016; GFSAD30m and Yang et al (2015) was compared with NLRB and reconstructed cropland area in 2015 and 1985, respectively; the black solid line is the 1:1 line and the colored dash lines are the linear regression lines).

---

## Referee Report (RR1)

Authors had updating the data according to suggestions of mine in last version and compared Xinjiang and Xizhang's data after adjusting the priority of satellite data. The explain are persuasive and updating improved the dependability of gridded data. There are also two suggestions for this version.

(1)For the last period of 1900 to 1948, this paper used the trends extracted from Yang et al.'s (2015b) cropland data from 1887, 1933, and 1952. It had better give the values of these years and estimation value of 1900 in the text.

(2) In the authors' response, this paper chose the PFT classification system for MODIS land cover product, in which cropland are defined as grid cells dominated by cultivated crops (>60%). By giving different priorities to each dataset, the weighting scores help reduce the possibility to allocate cropland to grids with low crop coverage. However, whether it would decrease the allocation or no allocation in non-cropland grid cells? For example, the results in some regions may have deviation on cropland coverage sourced from above dealing. Please explain this kind of uncertainty in discussion part.

---

## Author Response (AR2)

Authors had updating the data according to suggestions of mine in last version and compared Xinjiang and Xizhang's data after adjusting the priority of satellite data. The explain are persuasive and updating improved the dependability of gridded data. There are also two suggestions for this version.

Response: We thank the reviewer for the positive comments. We have made further revisions based on the suggestions. Please see our point-by-point reply to the comments below.

(1) For the last period of 1900 to 1948, this paper used the trends extracted from Yang et al.'s (2015b) cropland data from 1887, 1933, and 1952. It had better give the values of these years and estimation value of 1900 in the text.

Response: We thank the reviewer for the suggestion. We have added the descriptions in Lines292-295 and also added data in the Table 3 (Page 13) to show the cropland areas from Yang et al (2015) and our model simulations.

(2) In the authors' response, this paper chose the PFT classification system for MODIS land cover product, in which cropland are defined as grid cells dominated by cultivated crops (>60%). By giving different priorities to each dataset, the weighting scores help reduce the possibility to allocate cropland to grids with low crop coverage. However, whether it would decrease the allocation or no allocation in non-cropland grid cells? For example, the results in some regions may have deviation on cropland coverage sourced from above dealing. Please explain this kind of uncertainty in discussion part.

Response: We thank the reviewer for the suggestion. Yes, the approach also helps reduced the possibility to allocate cropland in non-cropland grid cells. This approach will not allocate cropland to grid cells if those grids were not classified as cropland in all data used. Indeed, the uncertainties maybe larger in some regions if the available data were majorly low-resolution images (with substantial mixed grid cells). We agree with the reviewer and added discussion about the uncertainty (see Lines412-417 in Page 19).